

# Mapping Individual Earthquake Preparedness in China

Guochun Wu[1], Ziqiang Han[2], Weijin Xu[1], Yue Gong[3]

[1] Institute of Geophysics, China Earthquake Administration, Beijing, 100081, China
[2] Institute for Disaster Management and Reconstruction, Sichuan University, Chengdu, 610207, China
[3] China Earthquake Disaster Prevention Center, Beijing, 100000, China

*Correspondence to*: Ziqiang Han (ziqiang.han@qq.com)

**Abstract.** Disaster preparedness is critical for reducing potential impact. This paper contributes to current knowledge of disaster preparedness using a representative national sample data from China, which faces high earthquake risks in many areas of the country. The adoption of earthquake preparedness activities by the general public, including five material

preparation, and five awareness preparation were surveyed, and 3,245 respondents from all of the 31 provinces of mainland China participated in the survey. Linear regression models and Logit regression models were used to analyze the effects of potential influencing factors. Overall, the preparedness levels are not satisfied, with a material preparation score of 3.02 (1-5), and awareness preparation score of 2.79 (1-5), nationally. Meanwhile, residents from west China where have higher earthquake risk have higher preparedness degrees. The concern of disaster risk reduction, the concern of building safety and

participation in public affairs are consistent positive predictors of both material and awareness preparedness. The demographic and socioeconomic variables' effects, such as gender, age, education, income, urban/rural division, and occupied building type, vary according to different preparedness activities. Finally, the paper concludes with a discussion of the theoretical contribution and potential implementation.

**Keywords**: earthquake; preparedness; China

## 1 Introduction

China is a country with high seismic risk. Within the last 100 years, one earthquake higher than Richer 7.5 degree (M≥7.5 in short) would occur every five years in China on average, and a M≥8.0 earthquake occurred about every 10 years. Though China only shares about 7% of the land area in the world, it has more than 35% of M≥7 continental earthquakes. 58% of the

land area, more than 50% of the cities and more than 70% of the urban population in China are actually residing in an area with high seismic risk---in the seismic zone with VII intensity degree or above (Gao et al., 2015).

Moreover, most parts of China are facing the threat of earthquakes. Though most of the recent earthquakes occurred in the western region, the east area with high population intensity is not totally free of threat. Based on the data from China Earthquake Network Centre (CENC), there were 130 earthquakes between M 6-7, 16 earthquakes between M 7-8 degree and

two earthquakes higher than M 8 degree occurred in Mainland China since 1980, and most of the M≥6.0 earthquake occurred



in western China and rarely occurred in the eastern area. Yunnan, Qinghai, Sichuan, Gansu, Xizang, and Xinjiang are prone-earthquake provinces. But when we look back for a longer time period, the east part of China also had many earthquakes in the history. From1500 to 1980, there were 94 M7-8 earthquakes and 15 earthquakes above M 8 degree in the mainland of China. Tancheng Earthquake (1698), Pinggu-Sanhe Earthquake (1679) and Tangshan Earthquake(1976)all occurred in

the north and eastern China, where has a large population (Figure 1). Thus, it can be concluded that seismic risk is a threat for most areas of China, and national studies covering all of China is needed.

[Figure 1 Here]

Pre-disaster mitigation and preparedness are key methods to reduce potential disaster impact. A prior study from the United States indicates that one dollar investment in pre-disaster mitigation and preparedness would reduce four dollars potential losses (Godschalk et al., 2009). Thus, preparedness becomes a research and practice priority in recent years. For example, a National Preparedness Strategy has been proposed in the United States, and prevention, protection, mitigation, response, and recovery are organized as the five mission areas of core capabilities of the National Preparedness Goal (FEMA,

2015). Preparedness is clearly stated as the "shared responsibility of all individuals, families, communities, private and nonprofit sectors, faith-based organizations, and levels of governments" (FEMA, 2016). Similarly, laws and regulations in People's Republic of China, such as the Earthquake Mitigation and Reduction Act, and the 2016-2020 National Comprehensive Disaster Risk Reduction Plan request more efforts on mitigation and preparedness, and the local government should take the responsibility of disaster preparedness education to increase the public's awareness, and to improve the

whole society's disaster response capacity (Anon, 2008; 国务院办公厅, 2016). Therefore, studying individual preparedness for disasters can provide valuable knowledge to disaster and emergency management practices, and ultimately reduce the disaster losses.

Theoretical models from varied research areas have been adopted by disaster preparedness related studies. The Protective Action Decision Model, Health Belief Model, Extended Parallel Process Model, Theory of Planned Behavior and

Social Cognitive Theories, Personal-Relative-To-Event Model are the commonly adopted research frameworks (Duval and Mulilis, 1999; Ejeta et al., 2015; Lindell and Perry, 2012). Overall, all these models follow the psychological-behavior pattern, but with different components, pathways and structures. Different terms are also used varied widely in literature. Jargons like protective behaviors/actions, hazards adjustment behaviors/actions, mitigation or preparedness are the commonly used to describe the individual and household's actions undertaken in anticipation of natural hazards (Bubeck et

al., 2012; Kohn et al., 2012; Lindell, 2013; Lindell and Perry, 2000; Wachinger et al., 2013). In this paper, the term "preparedness" is mainly used to describe these actions undertaken to keep consistency.

The attributes of natural hazards, the features of protective actions, and the perceived characteristics of related stakeholders are the three groups of interrelated determinants of household preparedness (Lindell, 2013).The features of protective/adjustment behaviors refer to the efficacy, safety, time requirement, perceived implementation barriers and cost of



undertaking that kind of preparedness action. For example, if an action needs special skills or very costly, many people would not adopt it. The stakeholder characteristics include the trust in varied stakeholders, feeling of responsibility etc. If one individual believes that the government agencies should take the main responsibility for disaster relief, that may reduce their motivation to adopt protective actions. The attributes of natural hazards cover the proximity to natural hazards, and

5 perceived risks, etc. Demographic and socioeconomic variables are included as controlled variables in most of the studies. Recent literature reviews indicate that the relationship between risk perception and household preparedness is hardly observed in empirical studies. The factors of coping appraisal, termed as the efficacy of preparedness actions by Lindell (Lindell, 2013) are consistently related to preparedness behaviors (Bubeck et al., 2012; Kohn et al., 2012). Some demographic (such as gender, income, education) or household characteristics (number of dependents in a household), trust

in stakeholder (government agencies in particular), previous disaster experience are predictors of household preparedness (Kohn et al., 2012). Overall the influencing factors of individual and household preparedness attitudes and behaviors are complex and multifaceted, and there is a need for further investigation.

For earthquake preparedness, in particular, demographic and socioeconomic variables are found to be predictors of adopting preparedness actions, however, they are not consistent (Lindell et al., 2009). An exploratory study from Istanbul

explored the association between earthquake preparedness and basic demographic variables like age, education, financial income, gender etc., only using cross-tabulated tables, and the results showed that earthquake preparedness in this region had minor variations (Eraybar et al., 2010). Lessons learned from Turkey exhibited the correlation between place of living, earthquake experience and preparedness actions (Oral et al., 2015). Another paper from Turkey as well indicated that risk components and characteristics, and socioeconomic variables were significant drivers of varied forms of mitigation actions

(Ozdemir and Yilmaz, 2011). The education level, living in a higher earthquake-prone area, participated in rescue and solidarity actions previously, knowledge, home ownership were significant predictors of preparedness in Istanbul (Tekeli-Yesil et al., 2010). Prior disaster experience and risk perception were found to be positive predictors of disaster preparedness in California (Han and Nigg, 2011), however, another survey on homeowners displayed that when the appraised threat increase, only those who had sufficient resources had significant higher earthquake preparedness (Duval and Mulilis, 1999).

Another questionnaire survey from Dhaka city of Bangladesh revealed that residential unit value and the individual's education level were positively influencing factors of the respondent's earthquake preparedness (Paul and Bhuiyan, 2010). Similar observation from Israel declared gender differences in earthquake risk perception and knowledge (Soffer et al., 2011). A qualitative study from New Zealand through the symbolic interactionism perspective demonstrated that how individual make meaning of earthquake information that they exposed to is related to their undertaking actual preparedness actions

(Becker et al., 2012). Another survey illustrated that the psychological factors like tendency to take risks and their locus of control, home ownership, and length of residence were significant predictors of earthquake preparation (Spittal et al., 2008). For earthquake proximity, Lindell and Prater's finding in the United States demonstrated that the ones living with high seismic hazard and another area of moderate seismic hazard did not show significant differences. Rather, the perception of





hazards-adjustment characteristics was correlated significantly with adoption intention and actual preparedness (Lindell and Prater, 2002).

Risk perception and people's preparedness behaviors vary across cultures and societies (Viklund, 2003). Within the Chinese cultural context, a prior analysis revealed that people having disaster experience (heavy-snow and earthquake in 2008) were not always more risk averse (Li et al., 2011). By comparing survey results from two cities with different smog exposures, Wei et.al. found that proximity to threat (smog) had little impact on individual's risk perception and protective behavior, though the participants from the two cities differ considerably in their smog experience (Wei et al., 2017). One survey of the survivors of the 2010 Yushu earthquake showed that individuals with a higher degree of trust in government would have lower self-reported preparedness degrees (Han et al., 2017). Another study from Taiwan indicated that prior earthquake experience mainly affected the perceived personal impact dimension of risk perception, but not the perceived controllability (sense of efficacy of self-protection) (Kung and Chen, 2012). Methods of risk communication may matter in encouraging individual's adaptation of preparedness actions. Psychology experiment result demonstrated that the ambiguity tolerance and source of information were interactive factors shaping people's risk perception and willingness to buy earthquake insurance. Participants with higher ambiguity tolerance felt riskier and were more likely to purchase earthquake insurance when risk message came from official sources rather than peers (Zhu et al., 2012). Unlike to limited studies exploring the individual and household's preparedness behaviors using a small sample from a specific geographical areas in China (Han et al., 2017; Wei et al., 2017), this paper analyzes the individual's earthquake preparedness using a representative national sample which would more precise in contributing current knowledge on individual and household' preparedness studies theoretically and practically.

This paper maps the individual earthquake preparedness in China using a national sample. It is organized as follows. First, the method (sampling, measurements, and data analysis strategy) are reported. Then, the association between earthquake proximity and preparedness are explored and presented using maps. Third, the effects of basic socioeconomic and demographic variables, the characteristics of building, degree of public participation, and risk perception on the overall preparedness and separated preparedness activities (stockpiling water, food, medicine, flash, radio; shelter awareness, participating drill, intention of purchasing insurance, telling the difference between prediction and warning, earthquake information seeking) are analyzed using varied regression models. Finally, the paper ends with a brief discussion of the theoretically and practically contribution, as well as future research directions.

## 2 Methods

### 2.1 Sampling

An online survey of earthquake reduction communication was conducted from September 21st to October 10th in 2015 by a professional marketing survey company, with the sampling requirement guidelines from the authors. Gender, age, and education status were controlled in the sampling process according to the 6th national population census data (NPCD). 100





samples in each province of the mainland China were planned to be surveyed with a 5% of the variance. After the survey, we made a random check of the respondents' URL to make sure that every respondent was unique. A total of 3245 participants from all 31 provinces in mainland China, and about 105 respondents from each province participated in the survey (Figure 2). Our sample was consistent with the 6th NPCD in terms of gender, age, with a little difference of education degrees (Table 1).

In our survey, 35% of the respondents had college and above education, but in the 6th NPCD, 20% of the population had attended college.

[Table 1 Here]

[Figure 2 Here]

## 2.2 Measurements

Preparedness activities: 10 preparedness activities were proposed in our survey, five were related with material stockpile within a household, and the other five were related to capacity building and participation. The question "In order to prepare for potential earthquakes, do you have the following materials stockpiled in your home?" was used in the survey. Water, food, medicine, flashlight, and radio were proposed. If the respondent chose "yes" to that kind of material preparedness, the variable was coded as one, otherwise, it was coded as zero. Meanwhile, the aggregation of the five material stockpile was

used as a material stockpile preparedness score, and thus, it became a continuous variable ranging from zero to five, indicating the increasing degree of the material stockpile.

Besides, we also inquired the respondent's other five preparedness related behaviors, termed as knowing emergency shelter nearby, having participated in emergency exercise/drills, the intention of purchasing earthquake insurance if available, knowing the difference between earthquake predicting and earthquake warning, having visited the China Earthquake

Administration Bureau's website or social media public communication page. If the respondent had positive feedback on one kind of the five activities, that variable was coded as one ("yes"), otherwise, it was coded as zero ("no"). At last, the sum of the 10 preparedness variables (five preparedness behaviors and five material stockpile) was generated as an overall degree of preparedness, ranging from zero to ten.

Influencing Factors: The respondents' occupied building characteristics, socioeconomic and demographic attributes,

and psychological variables were used to explore their effects on the preparedness. Meanwhile, the geographical variation at the provincial level was controlled in all the models but not reported in the tables. The building type captured the height of the buildings they occupied. It was categorized as one-story, two or three-story, four to six-story, or higher than seven-story. The age of the building they occupied was another variable used to measure the characteristics of the occupied buildings, and it was a continuous variable measured by years. Gender, age, education attainment were the demographic variables included.

Gender was a dummy variable, with one as male. Age was a continuous variable measured by years. Education was an ordinal variable ranking from one to five, representing the meaning of "Illiteracy or primary school", "Middle school", "High school", "College" and "Graduate or above". The annual income was measured by an ordinal variable ranking from one to three, meaning "less or equal to 60,000 RMB", "higher than 60,000 but less than 120,000 RMB" or "higher than





120,000 RMB". The rural-urban division was a dummy variable with one as an urban resident. We also included one measure of the respondent's participation in public affairs. It was obtained by the question "have you ever participated in your community vote?" and the answers were yes (1) or no (0). Two variables were adopted to capture the respondent's concern of safety. One was as "do you pay attention to the disaster risk reduction knowledge or issues during normal days?" and the answers were "Not at all (1)", "Not a lot (2)", "Neutral (3)", "Pay some attention (4)", "Pay lots of attention (5)". The other asked "Are you concern of your house safety?", and the answers were yes (1), and no (0).

## 2.3 Data Analysis

The 10 preparedness activities were categorized as materials preparedness (water, food, medicine, flashlight, radio) and awareness preparedness (knowing shelter, participating drill, the intention of purchasing insurance, knowing the difference between predict and warning, seeking information from the CEA's website or social media page). We first mapped the geographical distribution of the material preparedness and awareness preparedness scores using GIS. Then, the general regression models were adopted to explore the effects of the variables on material preparedness, and awareness preparedness, respectively. Lastly, we explored the effects of these influencing variables on each kind of the preparedness activities by logistic regression models. The statistical analysis was implemented by the statistical software Stata 13.1 MP version.

## 3 Results

The 3,245 respondents of our survey had an average age of 38.73. 46% of them were male, 61% were urban residents, 39% had participated in community vote before, 1.23% of them had primary school education attainment, 16.80% were middle school educated, 46.72% were high school educated, and 31.09% of them had attended college, and another 4.16% had graduate school education. 67.43% of them had an annual income less or equal than 60,000 RMB, 22.56% of them had an annual income between 60,000 to 120,000 RMB, and 10.01% earned more than 120,000 RMB each year. 11.98% of the respondents were living in the one-storey building, 22.53% of them were living in two or three-story building, 39.14% were in four to six-story building and 26.35% were in higher than seven-story buildings. 83% of the respondents concerned the safety of the buildings they occupied, about 68.51% of them indicated that they had paid attention to learning disaster risk reduction knowledge or skills.

In term of preparedness, 74% of the respondents had extra water stored at home, 72% of them had extra food, 65% had medicine in preparation, 69% had a flashlight at home, and 21% of them had radio prepared. 78% of them were aware that where was the nearest emergency shelter, 62% had participated in some kinds of emergency exercises or drills. If earthquake insurance was available, 41% of them would purchase. 45% of the respondents had visited the China Earthquake Administration bureau's website or social media (Weibo or Wechat) page for information. The aggregation of the five material-related preparedness activities was named as material preparedness in this paper, and it ranged from zero to five,





with an average value of 3.02, with a standard deviation of 1.57. The awareness preparedness (sum of the five awareness related actions) had a mean value of 2.79, with a standard deviation of 1.54 (Table 2).

[Table 2 Here]

## 3.1 Mapping the Preparedness Activities

The mean values of material preparedness (5 items) and awareness preparedness (5 items) by province were mapped in Figure 3 and Figure 4. The average score of material preparedness was 3.02, while the awareness preparedness score was 2.79, both with a range from one to five. Overall, respondents in the western China, where had higher earthquake risks, had higher preparedness score. In terms of material preparation, the top five provinces were Yunnan (3.45), Qinghai (3.4), Fujian (3.38), Guizhou (3.36), and Sichuan (3.28), while the least three prepared provinces were Hunan (2.6), Hubei (2.7) and Henan (2.71). For awareness preparedness, the top five prepared provinces were Yunnan (3.31), Sichuan (3.27), Xizang (3.27), Gansu (3.26) and Guizhou (3.26), while Shanghai (2.15), Beijing (2.17), Jiangsu(2.29) , Hebei(2.39) and Hubei(2.43) were the five least prepared.

[Figure 3 Here]

[Figure 4 Here]

## 3.2 Influencing Factors of Preparedness Behaviours

We first regressed on the awareness preparedness score and material preparedness score using general linear regression models. The adjusted R2 for the awareness preparedness model was 0.332 while the adjusted R2 for the material preparedness was 0.110. Overall, the psychological factors and participation variables were positive predictors of preparedness. With a higher degree of concern for building safety and concern for disaster risk reduction, the respondents would have a higher degree of both awareness preparedness and material preparedness. The ones who had participated in community voting would also have both higher degrees of awareness and material preparedness compared with the ones who had never participated in the voting. Being male was also positively associated with both awareness and material preparedness. The elders would have a lower degree of awareness preparedness, but such difference on material preparedness was not significant. Annual income was also positively correlated with awareness preparedness, but not material preparedness. It's out of our expectation that urban residents had lower awareness preparedness and material preparedness degrees, though such effect on awareness preparedness was not statistically significant. The type of buildings (height) did not affect the awareness preparedness, but people living in higher story buildings would prepare more materials. The building age's effect was not significant in predicting the material preparedness but was negatively associated with awareness preparedness (Table 3).

[Table 3 Here]



[Table 4 Here]

[Table 5 Here]

The impact of the proposed predictors on each kind of material preparedness and awareness preparedness were estimated

using logistic regression models, and the results (odds ratios) were reported in Table 4 and Table 5. Overall, the concern of building safety and concern of disaster risk reduction were the two most consistent and strongest positive predictors of almost all the ten preparedness behaviors, besides the insignificant effect of the concern of building safety on knowing nearby shelters. The participation variable (voting) were strong predictors of all the five awareness preparedness actions, but its effects on most of the material preparedness, such as water, food, flashlight, and radio were not statistically significant.

Being a male was significantly more possible to obtain a radio, know the nearby shelter, and tell the difference between earthquake warning and predicting. The elders didn't demonstrate significant differences in all the five material preparing, but they would have a slightly lower probability of participating a drill, purchasing insurances, telling the difference between predict and warning, and seeking earthquake-related information. The education was significantly positively associated with participating a drill and preparing water and food at home. The annual income was only significantly correlated with higher

probability of preparing medicine at home, purchasing insurance, and seeking earthquake-related information. The urban residents had a significant lower probability of preparing food, water, and medicine at home compared with rural residents, and they would also have a lower probability of participating emergency drills. The building type and age of the occupied buildings' effects were not significant for most of the preparedness activities.

## 4 Discussion

In this paper, we analyze the individual's preparedness activities for the earthquake in China using a national sample. We find that the public in western China, where has higher seismic risks, do have a higher degree of preparedness, for both material preparedness and awareness preparedness. Most of the least prepared are in the eastern provinces. This indicates that most of the public is aware of the earthquake risk in their region, generally. This result also demonstrates that hazards proximity is positively correlated with hazards (earthquake) preparedness (Bonaiuto et al., 2016; Howe, 2011; Lindell, 2013;

Mishra et al., 2010; Russell et al., 1995; Zhang et al., 2010).

We differentiate the preparedness activities into material preparedness and awareness preparedness. Overall, our data show that the concern of disaster risk reduction and concern of building safety are positively associated with both material preparedness and awareness preparedness. Moreover, the effects of concern of disaster risk reduction are positive with all the five material preparedness activities and the five awareness activities. The concern with building safety's positive

effects is not significant for the "knowing shelter" only. The concern of disaster risk reduction and concern of building safety can be seen as risk perception. Similar to most of the prior studies, risk perception is a positive predictor of individual





disaster preparedness (Bronfman et al., 2016; Han et al., 2017; Han and Nigg, 2011; Sadiq and Graham, 2016; Zhang et al., 2010).

The participation of public affairs (vote) is significantly and positively associated with the overall awareness preparedness score and separated awareness preparedness activities, but most of such correlations with individual material
preparedness are not significant, thought the association with the overall material preparedness score is significant. In this paper, we innovatively explored the role of public participation in individual disaster preparedness. Prior studies have demonstrated that trust in relevant stakeholders, such as trust in government could actually discourage individual's preparedness (Han et al., 2017; Terpstra, 2011), though some studies provide reverse or non-significant evidence (Basolo et al., 2009; DeYoung et al., 2016). Moreover, the feeling of responsibility---when an individual feels more responsible for
personal safety, they would prepare more for potential hazards (Arceneaux and Stein, 2006; Mulilis and Duval, 1997; Wei et al., 2017). Our results demonstrate that individual's participation in general public affairs could be a good predictor of individual's disaster preparedness because disaster is more like a public issue that would impact both individuals and the public, and also a shared responsibility between individuals and public.

In sum, we significantly contribute to current disaster preparedness studies by using a national data from China,
exploring the role of public participation, and concern of building safety, as well as the concern of disaster risk reduction. But this paper does have two limitations. First, we only explored the variations of preparedness at province level, which is quite large and blur. Future studies with more specific geographical locations which can measure the proximity to hazards are needed. Second, we did not include the efficacy (Roush and Tyson, 2012; Samaddar et al., 2014) of the preparedness activities in our analysis, and the covering of these factors do need in future.

**5 Conclusions**

This paper maps the earthquake preparedness in mainland China using a representative national sample, by the first time as we know. Ten earthquake preparedness activities are proposed, five of them are material preparation, and five of them are awareness preparation. Overall the preparedness degrees are not satisfied, with a national material preparedness score of 3.02 (1-5), and a national awareness preparedness score of 2.79 (1-5). In terms of geographical distribution, the western China
where has experienced recently earthquakes has relatively higher degrees of preparation, for both material and awareness preparedness. The concern of disaster risk reduction, the concern of building safety, and participation in public affairs (vote) are consistent positive predictors of both material preparedness and awareness preparedness. The role of gender, age, education, income, urban/rural divisions, and occupied building characteristics vary according to different preparedness activities.





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





**Figure 1: Historical Earthquakes in Mainland China from 1500.**

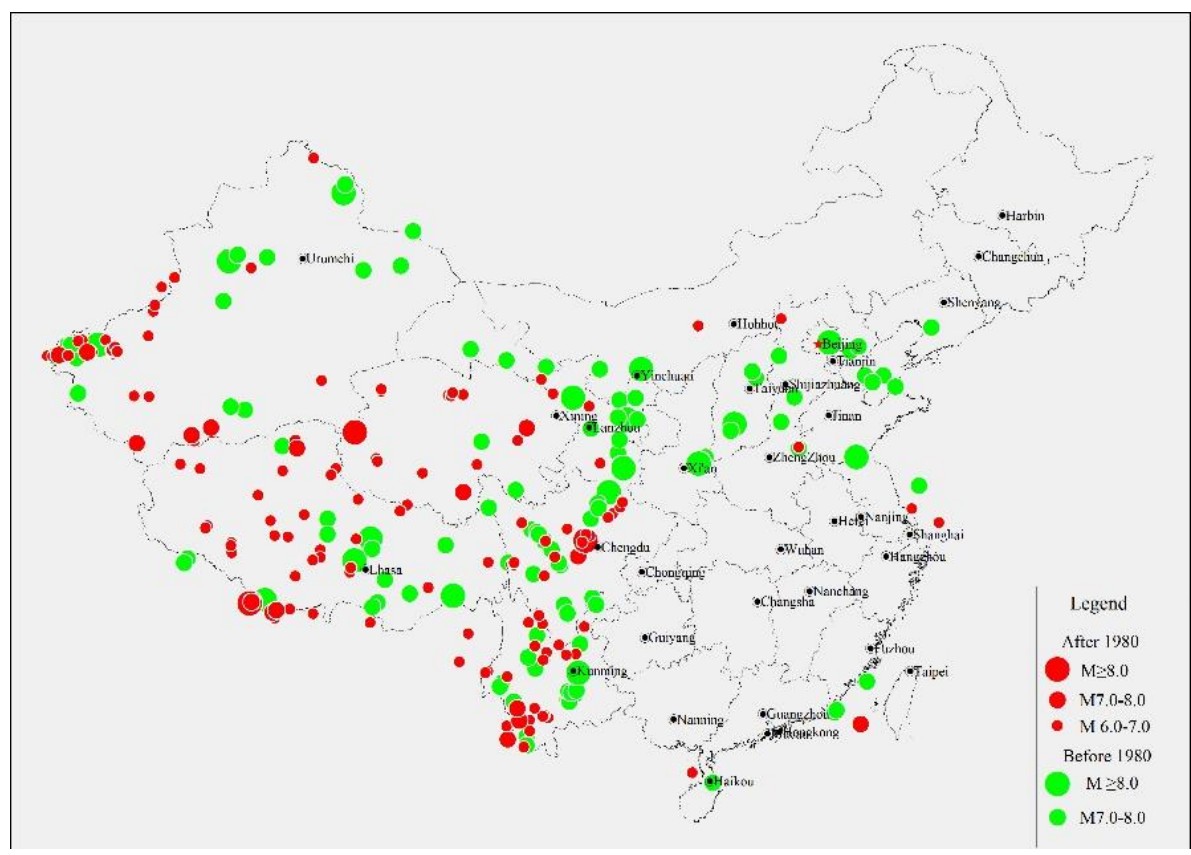

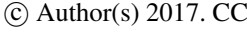



**Figure 2: Geographical Distribution of the Sample.**

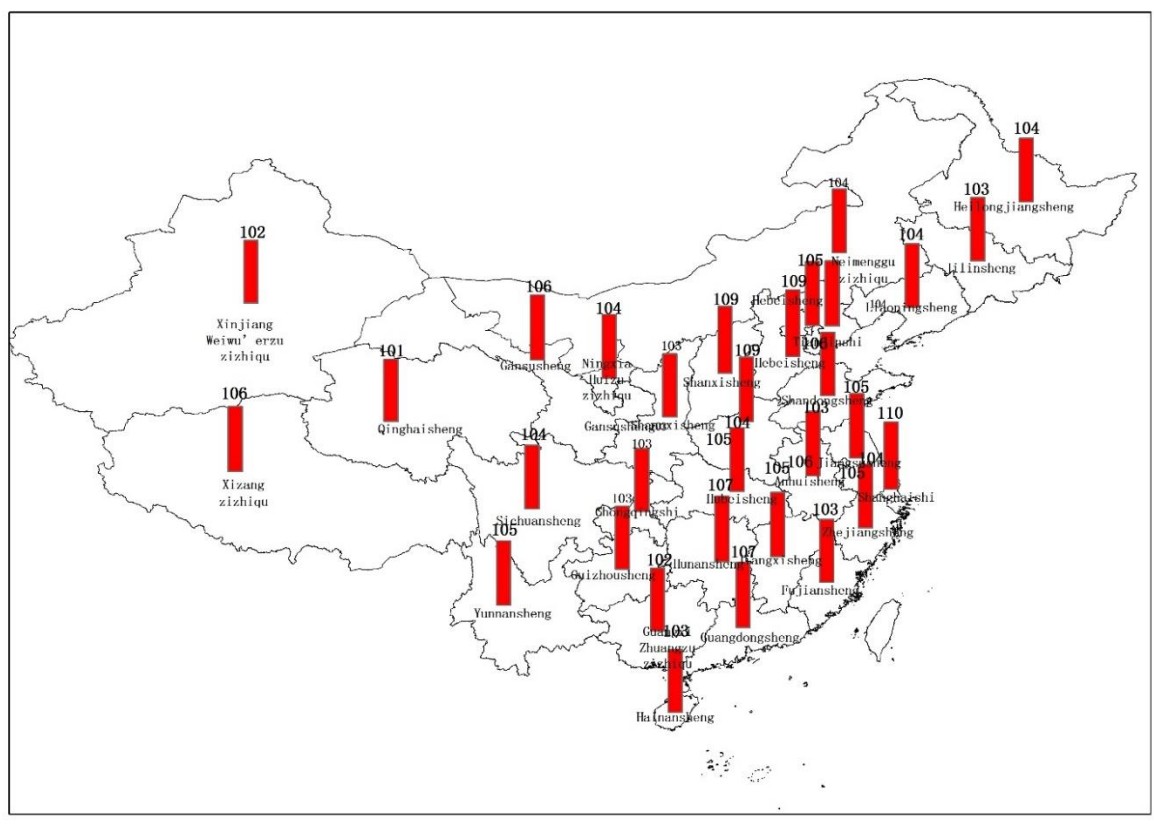




**Figure 3:  Mean Value of Material Preparedness.**

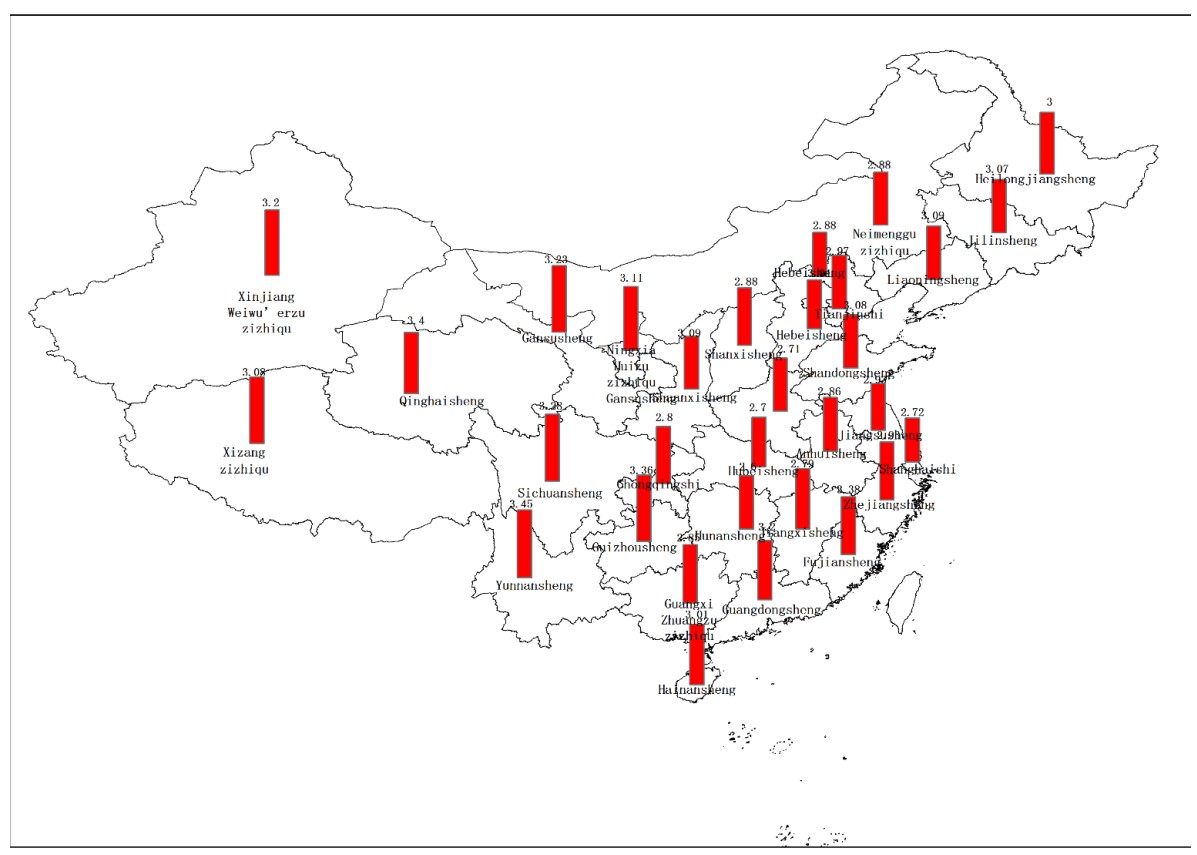



**Figure 4: Mean Value of Awareness Preparedness.**

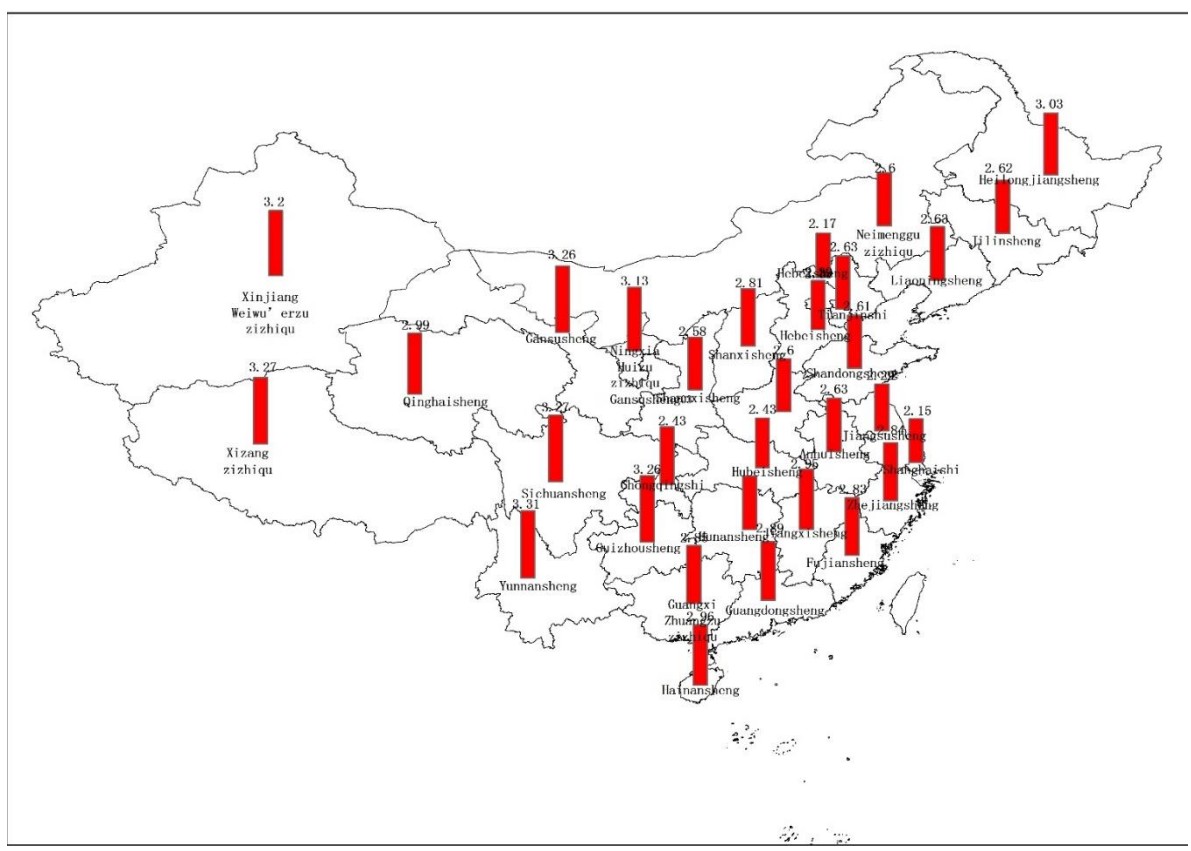



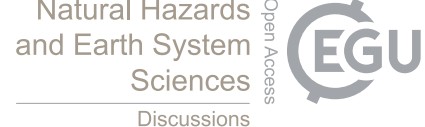

**Table 1.** Comparison of Sample and the National Population

|  |  | 6th NPCD | Survey result |
| --- | --- | --- | --- |
| Gender | Male | 50% | 46.4% |
|  | Female | 50% | 53.6% |
| Age | Under 18 | 10% | 7% |
|  | 19-29 | 25% | 27.1% |
|  | 30-39 | 21% | 22.1% |
|  | 40-49 | 15% | 14.2% |
|  | 50-59 | 14% | 14.8% |
|  | Over 60 | 15% | 14.9% |
| Education | College and above | 20% | 35.3% |





**Table 2.** Descriptive Analysis

| Variable | Mean | SD | Min | Max | Variable | | Frequency | Percent |
|---|---|---|---|---|---|---|---|---|
| Prepare Awareness | 2.79 | 1.54 | 0 | 5 | Education | *Primary or lower* | 40 | 1.23 |
| Prepare Material | 3.02 | 1.57 | 0 | 5 | *Middle School* | | 545 | 16.80 |
| Water | 0.74 | 0.44 | 0 | 1 | *High School* | | 1,516 | 46.72 |
| Food | 0.72 | 0.45 | 0 | 1 | *College* | | 1,009 | 31.09 |
| Medicine | 0.65 | 0.48 | 0 | 1 | *Graduate or above* | | 135 | 4.16 |
| Flash | 0.69 | 0.46 | 0 | 1 | Income Category | <60,000 | 2,188 | 67.43 |
| Radio | 0.21 | 0.41 | 0 | 1 | *(60,000-120,000]* | | 732 | 22.56 |
| Shelter | 0.78 | 0.42 | 0 | 1 | *>120,000* | | 325 | 10.01 |
| Drill | 0.62 | 0.49 | 0 | 1 | Building Type | *One-storey* | 389 | 11.98 |
| Insurance | 0.41 | 0.49 | 0 | 1 | *2-3-storey* | | 731 | 22.53 |
| Seek Info | 0.45 | 0.50 | 0 | 1 | *4-6 storey* | | 1,270 | 39.14 |
| Male | 0.46 | 0.50 | 0 | 1 | *Higher than 7-storey* | | 855 | 26.35 |
| Age | 38.73 | 15.93 | 15 | 68 | Concern of DRR | *Not at all* | 14 | 0.43 |
| Urban | 0.61 | 0.49 | 0 | 1 | *Not very concern* | | 146 | 4.50 |
| House Age | 11.50 | 10.85 | 0.20 | 65 | *Neutral* | | 862 | 26.56 |
| Vote | 0.39 | 0.49 | 0 | 1 | *Concern some* | | 1,514 | 46.66 |
| Concern building safety | 0.83 | 0.37 | 0 | 1 | *Very concern* | | 709 | 21.85 |
| | | | | | Total | | 3,245 | 100 |





**Table 3.** Regression on Material Preparedness and Awareness Ready (N=3,245)

|  | Awareness Preparedness | Material Preparedness |
|---|---|---|
| Male | 0.13** | 0.12* |
|  | (0.05) | (0.05) |
| Age | -0.02*** | -0.00 |
|  | (0.00) | (0.00) |
| Education | 0.01 | 0.09* |
|  | (0.04) | (0.04) |
| Income | 0.20*** | 0.05 |
|  | (0.05) | (0.05) |
| Urban | -0.10 | -0.13* |
|  | (0.05) | (0.06) |
| Building type    2-3-storey | 0.02 | 0.13 |
|  | (0.08) | (0.10) |
| *4-6 storey* | -0.02 | 0.21* |
|  | (0.08) | (0.09) |
| *Higher than 7-storey* | 0.01 | 0.27* |
|  | (0.09) | (0.10) |
| Building Age | -0.01** | -0.00 |
|  | (0.00) | (0.00) |
| Vote | 0.63*** | 0.13* |
|  | (0.05) | (0.06) |
| Concern of DRR | 0.65*** | 0.28*** |
|  | (0.03) | (0.04) |
| Concern of building safety | 0.62*** | 0.79*** |
|  | (0.06) | (0.07) |
| $R^2$ | 0.332 | 0.110 |

Standard errors in parentheses; * $p < 0.05$, ** $p < 0.01$, *** $p < 0.001$; the geographical variations were controlled at provincial level but not reported in the table.

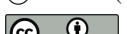



**Table 4.** Logistic Regression on Material Preparedness by Type (N=3,245)

| | | Water | Food | Medicine | Flash | Radio |
|---|---|---|---|---|---|---|
| Male | | 1.00 | 1.13 | 1.10 | 1.03 | 1.53*** |
| | | (0.09) | (0.10) | (0.09) | (0.08) | (0.14) |
| Age | | 0.99 | 1.00 | 1.00 | 1.00 | 1.00 |
| | | (0.00) | (0.00) | (0.00) | (0.00) | (0.00) |
| Education | | 1.23** | 1.16* | 1.05 | 1.04 | 1.00 |
| | | (0.08) | (0.07) | (0.06) | (0.06) | (0.07) |
| Income | | 0.90 | 0.99 | 1.18* | 1.10 | 1.15 |
| | | (0.08) | (0.08) | (0.09) | (0.09) | (0.10) |
| Urban | | 0.75** | 0.81* | 0.76** | 0.94 | 1.23 |
| | | (0.08) | (0.08) | (0.07) | (0.09) | (0.14) |
| Building type | *2-3-storey* | 0.98 | 1.16 | 1.30 | 1.27 | 0.96 |
| | | (0.15) | (0.17) | (0.18) | (0.18) | (0.16) |
| | *4-6 storey* | 1.13 | 1.22 | 1.49** | 1.33* | 1.02 |
| | | (0.17) | (0.18) | (0.20) | (0.18) | (0.17) |
| | *Higher than 7-storey* | 1.31 | 1.40* | 1.58** | 1.40* | 0.91 |
| | | (0.22) | (0.23) | (0.24) | (0.21) | (0.16) |
| Building Age | | 1.00 | 1.00 | 0.99 | 1.00 | 1.01 |
| | | (0.00) | (0.00) | (0.00) | (0.00) | (0.00) |
| Vote | | 1.09 | 1.14 | 1.37*** | 1.05 | 1.12 |
| | | (0.10) | (0.10) | (0.12) | (0.09) | (0.10) |
| Concern of DRR | | 1.45*** | 1.37*** | 1.34*** | 1.34*** | 1.23*** |
| | | (0.08) | (0.07) | (0.07) | (0.07) | (0.07) |
| Concern of building safety | | 2.50*** | 2.16*** | 2.22*** | 2.15*** | 1.57** |
| | | (0.27) | (0.23) | (0.23) | (0.22) | (0.22) |
| Pseudo $R^2$ | | 0.071 | 0.059 | 0.059 | 0.049 | 0.042 |

Odds Ratios were reported; Standard errors in parentheses; * $p < 0.05$, ** $p < 0.01$, *** $p < 0.001$; the geographical variations were controlled at provincial level but not reported in the table.





**Table 5.** Logistic Regression on Preparedness Awareness (N=3,245)

| | | Shelter | Drill | Insurance | Predict | Seek info |
|---|---|---|---|---|---|---|
| Male | | 1.32** | 0.87 | 1.13 | 1.32*** | 1.16 |
| | | (0.12) | (0.07) | (0.09) | (0.11) | (0.10) |
| Age | | 1.00 | 0.96*** | 0.99*** | 0.98*** | 0.99*** |
| | | (0.00) | (0.00) | (0.00) | (0.00) | (0.00) |
| Education | | 1.00 | 1.16* | 0.95 | 1.04 | 0.91 |
| | | (0.07) | (0.08) | (0.06) | (0.06) | (0.06) |
| Income | | 1.07 | 1.00 | 1.52*** | 1.13 | 1.42*** |
| | | (0.10) | (0.08) | (0.12) | (0.09) | (0.12) |
| Urban | | 0.87 | 0.66*** | 1.02 | 1.03 | 0.97 |
| | | (0.09) | (0.07) | (0.10) | (0.10) | (0.09) |
| Building type | *2-3-storey* | 0.84 | 1.21 | 0.97 | 1.09 | 1.02 |
| | | (0.14) | (0.19) | (0.14) | (0.15) | (0.15) |
| | *4-6 storey* | 0.77 | 0.86 | 1.03 | 1.21 | 1.00 |
| | | (0.12) | (0.13) | (0.15) | (0.17) | (0.14) |
| | *Higher than 7-storey* | 0.78 | 0.92 | 0.91 | 1.30 | 1.10 |
| | | (0.14) | (0.15) | (0.15) | (0.20) | (0.17) |
| Building Age | | 1.00 | 0.99* | 1.00 | 0.99 | 0.99*** |
| | | (0.00) | (0.00) | (0.00) | (0.00) | (0.00) |
| Vote | | 1.62*** | 2.25*** | 1.67*** | 1.90*** | 2.18*** |
| | | (0.16) | (0.20) | (0.14) | (0.16) | (0.18) |
| Concern of DRR | | 1.72*** | 1.72*** | 2.27*** | 2.03*** | 2.20*** |
| | | (0.10) | (0.10) | (0.13) | (0.11) | (0.13) |
| Concern of building safety | | 1.07 | 1.59*** | 3.99*** | 1.86*** | 2.64*** |
| | | (0.12) | (0.18) | (0.59) | (0.21) | (0.34) |
| Pseudo $R^2$ | | 0.071 | 0.180 | 0.163 | 0.121 | 0.170 |

Odds Ratios were reported; Standard errors in parentheses; * $p < 0.05$, ** $p < 0.01$, *** $p < 0.001$; the geographical variations were controlled at provincial level but not reported in the table.