# Peer review of "Mapping Individual Earthquake Preparedness in China"

_Natural Hazards and Earth System Sciences, 2017_

## Referee Comment (RC1) · Anonymous Referee #1 · 27 Dec 2017

This paper explored the individual earthquake preparedness behavior in China using a national sample. It will add valuable knowledge to current understanding of earthquake mitigation and preparedness. The research methods and results are appropriate, and well presented. However, I do have several concerns and suggestions to make this paper better. I suggest to accept this paper with minor revision. If the authors can address my concerns appropriately, I hope no need to review it again. 1. Few type errors, like page 2, line 20, the citation is in Chinese. 2. The introduction section, especially the paragraphs in page 3 and page 4, can be written more concisely. 3. The authors should clearly state their research questions or hypothesis in the end of the introduction section. 4. The discussion section could be more interactive. How this study could enhance the earthquake mitigation and preparedness practice in China

and worldwide should be discussed with several sentences in the discussion section. 5. When the authors discussed the limitations, I think there are more things can do on this topic in future since there are not many studies on this topic from China yet. Thus, I would like to see one or two more discussions on the research limitations, and potential future research directions.

---

## Referee Comment (RC2) · Anonymous Referee #2 · 15 Mar 2018

Although the statistics used are valid - it is not clear what the research questions are. These should be clear and then the results should be presented in the order of these questions.

Was there a power analysis conducted prior to data collection? Please clarify.

Finally, there are also some minor technical issues that need to be addressed (for example, use of contractions in the paper).
* * *

---

## Short Comment (SC1) · 15 Mar 2018

Basically, this paper is interesting. Especially the questionnaire data from almost all the provinces of Chine, sample following the population structure, are valuable and one of the results that participation in public issues is positively related to preparedness is quite interesting even in the socio-political context of China. I think that this paper could be accepted with some minor revisions and that to make this paper better the followings should/could be revised: (1) the novelty of this paper is not necessarily clear even though it can be understood easily. Not only uniqueness of the data but the novelty of question(s) should be written more clearly in the introductory section in relation to the purpose of this paper. (2) The literature review is very much convenient for understanding the research trends, but it should be more focused on the topics directly

related to the purpose of this paper. (3) For Fig. 3 and Fig. 4, it is better to use a choropleth map, not a chart map (bar chart). And, if possible, a brief explanation could be added in Section 3.1 (p.7) referring to Fig. 1. (4) It might be desired to put detail explanations in the discussion part, about why public participation is related to preparedness in the context of China, referring to some social theories. (5) Discussion of the paper's limitation should be moved to the concluding section.

---

## Author Comment (AC1) · 26 Mar 2018

Nat. Hazards Earth Syst. Sci. Mapping Individual Earthquake Preparedness in China (nhess-2017-391)

Authors' Responses to the Reviewers' Comments

We thank the anonymous reviewer for your constructive feedback. We have addressed all your concerns, incorporated your suggestions, and below is a detailed memo documenting the changes we made to the manuscript. Please note that we only answered the negative comments/concerns in this memo. The changes in the manuscript are highlighted in yellow.

Comment 1: This paper explored the individual earthquake preparedness behavior in

China using a national sample. It will add valuable knowledge to current understanding of earthquake mitigation and preparedness. The research methods and results are appropriate, and well presented. However, I do have several concerns and suggestions to make this paper better. I suggest to accept this paper with minor revision. If the authors can address my concerns appropriately, I hope no need to review it again. 1. Few type errors, like page 2, line 20, the citation is in Chinese.

Authors' Response: Thanks very much for this correction, we have changed the citation information into English.

Comment 2: The introduction section, especially the paragraphs in page 3 and page 4, can be written more concisely.

Authors' Response: Thanks very much for this constructive suggestion. We have re-written some of the sentences, and deleted the unnecessary words to make these two paragraphs more concisely.

Comment 3: The authors should clearly state their research questions or hypothesis in the end of the introduction section.

Authors' Response: Thanks for the reviewer's kindness comment. We have re-written the last paragraph of the Introduction section by clearly stating our research questions as follows: "By analyzing this national representative sample, we characterized the individual's earthquake preparedness in China. In detail, the central questions of concern are: (1) will residents in the west of China (proximity to earthquake) have higher degrees of preparedness in general? (2) Would people with higher risk perceptions to an earthquake (e.g., the concern of disaster risk reduction and the concern of building safety) have a higher degree to preparedness; and (3) is participation in public affairs associated with higher degrees of earthquake preparedness? Besides the national representativeness of the data, we novelly explored the correlation between public involvement and the adoptions of disaster preparedness activities in China.

Comment 4: The discussion section could be more interactive. How this study could enhance the earthquake mitigation and preparedness practice in China and worldwide should be discussed with several sentences in the discussion section.

Authors' Response: Thanks for the reviewer's constructive comment. We have added one sentence in the last paragraph of the discussion section: "The findings of this paper also provide valuable implications for disaster risk reduction practice: people with higher degrees of participation in public affairs would also like to invest more in disaster preparedness. The involvement in disaster risk reduction activities cannot be separated from the involvement in other public issues."

Comment 5: 5. When the authors discussed the limitations, I think there are more things can do on this topic in future since there are not many studies on this topic from China yet. Thus, I would like to see one or two more discussions on the research limitations, and potential future research directions.

Authors' Response: Thanks for the constructive suggestion. We have added one more limitation: "Third, the preparedness at organizational and community level should be investigated as well."

---

## Author Comment (AC2) · 26 Mar 2018

Nat. Hazards Earth Syst. Sci. Mapping Individual Earthquake Preparedness in China (nhess-2017-391)

Authors' Responses to the Reviewers' Comments

We thank the anonymous reviewer for your constructive feedback. We have addressed all your concerns, incorporated your suggestions, and below is a detailed memo documenting the changes we made to the manuscript. Please note that we only answered the negative comments/concerns in this memo. The changes in the manuscript are highlighted in yellow. Comment 1: Although the statistics used are valid - it is not clear what the research questions are. These should be clear and then the results should

be presented in the order of these questions.

Authors' Response: Thanks very much for the constructive comment. We have rewritten the last paragraph of the Introduction section, and stated our research questions clearly. The results were presented in the order of the research questions. The last paragraph of the Introduction section is as following: "By analyzing this national representative sample, we characterized the individual's earthquake preparedness in China. In detail, the central questions of concern are: (1) will residents in the west of China (proximity to earthquake) have higher degrees of preparedness in general? (2) Would people with higher risk perceptions to an earthquake (e.g., the concern of disaster risk reduction and the concern of building safety) have a higher degree to preparedness; and (3) is participation in public affairs associated with higher degrees of earthquake preparedness? Besides the national representativeness of the data, we novelly explored the correlation between public involvement and the adoptions of disaster preparedness activities in China."

Comment 2: Was there a power analysis conducted prior to data collection? Please clarify.

Authors' Response: We appreciate this comment. Honestly, we did not conduct power analysis prior to data collection, because we planned to collect a large sample data, and thus the number of observations would be and actually is much larger than the minimum needs.

Comment 3: Finally, there are also some minor technical issues that need to be addressed (for example, use of contractions in the paper).

Authors' Response: Thanks for this comment. We have addressed all the contractions issues and other technical issues, reporting the full names when they appeared at the first time.

2017-391, 2017.

---

## Author Comment (AC3) · 26 Mar 2018

Nat. Hazards Earth Syst. Sci. Mapping Individual Earthquake Preparedness in China (nhess-2017-391)

Authors' Responses to the Reviewers' Comments

We thank the reviewer for your constructive feedback. We have addressed all your concerns, incorporated your suggestions, and below is a detailed memo documenting the changes we made to the manuscript. Please note that we only answered the negative comments/concerns in this memo. The changes in the manuscript are highlighted in yellow.

Comment 1: Basically, this paper is interesting. Especially the questionnaire data

from almost all the provinces of Chine, sample following the population structure, are valuable and one of the results that participation in public issues is positively related to preparedness is quite interesting even in the socio-political context of China. I think that this paper could be accepted with some minor revisions and that to make this paper better the followings should/could be revised: (1) the novelty of this paper is not necessarily clear even though it can be understood easily. Not only uniqueness of the data but the novelty of question(s) should be written more clearly in the introductory section in relation to the purpose of this paper.

Authors' Response: Thanks a lot for reminding us of this important point. We have rewritten the last paragraph of the introduction section, to make the novelty of the paper and the research questions clearer. "By analyzing this national representative sample, we characterized the individual's earthquake preparedness in China. In detail, the central questions of concern are: (1) will residents in the west of China (proximity to earthquake) have higher degrees of preparedness in general? (2) Would people with higher risk perceptions to an earthquake (e.g., the concern of disaster risk reduction and the concern of building safety) have a higher degree to preparedness; and (3) is participation in public affairs associated with higher degrees of earthquake preparedness? Besides the national representativeness of the data, we novelly explored the correlation between public involvement and the adoptions of disaster preparedness activities in China."

Comment 2: (2) The literature review is very much convenient for understanding the research trends, but it should be more focused on the topics directly related to the purpose of this paper.

Authors' Response: Thanks very much for this constructive suggestion. We have rewritten the introduction section, especially the paragraph 3 and 4 to make the literature review more concise and directly related.

Comment 3: (3) For Fig. 3 and Fig. 4, it is better to use a choropleth map, not a chart

map (bar chart). And, if possible, a brief explanation could be added in Section 3.1 (p.7) referring to Fig. 1.

Authors' Response: Thanks for the reviewer's kindness comment. We have replaced the Fig.3 and Fig.4 using choropleth maps. We also added one brief explanation refereeing to Fig.1 at the end of Section 3.1. "Compared to the historical earthquake records in China (Fig. 1), the people in the west of China, where have more earthquake records had higher degree of preparedness."

Comment 4: (4) It might be desired to put detail explanations in the discussion part, about why public participation is related to preparedness in the context of China, referring to some social theories.

Authors' Response: Thanks for the reviewer's constructive comment. We have added more discussion about public participation and the disaster preparedness in the discussion section.

Comment 5: (5) Discussion of the paper's limitation should be moved to the concluding section. Authors' Response: Thanks for the reviewer's suggestion. We noticed that the many relevant articles published in this journal had the limitation in discussion section. So we would like to keep it in the discussion section rather than the conclusion section. But we would like to communicate with the reviewer for potential change if the reviewer insisted.

---

## Author Response (AR2)

Dear Thomas Glade

Thanks very much for all your help. Here are our responses to your comments.

**Comment 1**: *"+ Please provide for ALL maps either coordinates or a north arrow and a scale bar."*

Response: Thanks for the comment. We have added the north arrow and scale bar in all maps.

**Comment 2:** "*+ Fig. 2: The title is not clear. Please provide a more suitable one, which explains as well the bars in the different regions."*

Response: Thanks very much. We have changed the expression of the title as "The Frequency Distribution of the Sample by Province."

**Comment 3:** "*+ Tab. 1: since "%" is the uniut for both last columns, please delete it within the table and put it to the column heading.*"

Response: We appreciate the comment. We have changed the Tab.1 as the editor suggested.

**Comment 4:** "*+ Tab 2: Are the two decimals within the columns "mean", "SD" and "Percent" really necessary? I would definitely delete the second decimal, if not both.*"

Response: We appreciate the comment. Since all the other numbers are two decimals, we hope we can keep the two decimals to keep all the manuscript consistence. But if the editor insists that it's not necessary to keep the two decimals, the editor can just keep one decimal on behalf of the authors. We have no disagreement on this.

**We noticed that the maps in the pdf file are not very clear. Thus, we submitted all the maps as supplement materials. The editor can use these maps instead if needed.**